# VIDEO PREDICTION USING SCORE-BASED CONDITIONAL DENSITY ESTIMATION

## ABSTRACT

Temporal prediction is inherently uncertain, but representing the ambiguity in natural image sequences is a challenging high-dimensional probabilistic inference problem. For natural scenes, the curse of dimensionality renders explicit density estimation statistically and computationally intractable. Here, we describe an implicit regression-based framework for learning and sampling the conditional density of the next frame in a video given previous observed frames. We show that sequence-to-image deep networks trained on a simple resilience-to-noise objective function extract adaptive representations for temporal prediction. Synthetic experiments demonstrate that this score-based framework can handle occlusion boundaries: unlike classical methods that average over bifurcating temporal trajectories, it chooses among likely trajectories, selecting more probable options with higher frequency. Furthermore, analysis of networks trained on natural image sequences reveals that the representation automatically weights predictive evidence by its reliability, which is a hallmark of statistical inference[1].

## 1 INTRODUCTION

All organisms make temporal predictions, and more complex organisms make prediction about more complex aspects of their environment. Even in the simplest cases and at very short timescales, temporal prediction is difficult: sensory measurements are incomplete and insufficient to fully specify the future, making prediction uncertain. Consider for example observing an image sequence: the next frame in that sequence is only incompletely determined by past frames, and as a result there is a whole distribution of possible next-frames. Typically, the distribution of the next-frame has more than one mode, making video prediction challenging. In particular, the classical Minimum Mean Squared Error approach to prediction (MMSE, Wiener, 1942) produces inaccurate and blurry solutions. This failure is well understood: the MMSE solution is given by the expectation of the next-frame, but since the distribution of the next-frame is multimodal, its expectation tends to have a low probability. To avoid blurry predictions, video prediction methods based on optic flow have been developed in image processing, and form the basis of most commonly used video codecs. Unfortunately, these flow-based methods are prone to errors when motion is discontinuous, *e.g.*, at occlusion boundaries, or non-rigid, *e.g.*, on deformable objects. These failures highlight the limitations of deterministic approaches: point estimates are inadequate for handling ambiguous situations. We will consider a probabilistic formulation of video prediction: estimating and sampling from the conditional density of the next frame given the previous $\tau$ frames in the image sequence, $p(x_{t+1}|x_t, \ldots, x_{t-\tau+1})$. Abstractly, temporal prediction can be thought of as an inverse problem that requires prior information in order to recover the part of the image sequence obfuscated by the arrow of time.

Learning a density from data is generally considered intractable for high-dimensional signals such as videos: the sample complexity of density estimation is exponential in the signal dimensionality (the so-called "curse of dimensionality"). As a result, the statistical difficulty of high-dimensional density modeling can only be approached by making strong assumptions, and many methods have been proposed to tackle the case of video prediction (Oprea et al., 2020). Each of these methods imposes inductive biases through choices of objective function and/or prediction architecture. But, overall, video prediction remained largely out of reach for statistical machine learning methods until

---

[1]Code will be released upon acceptance.

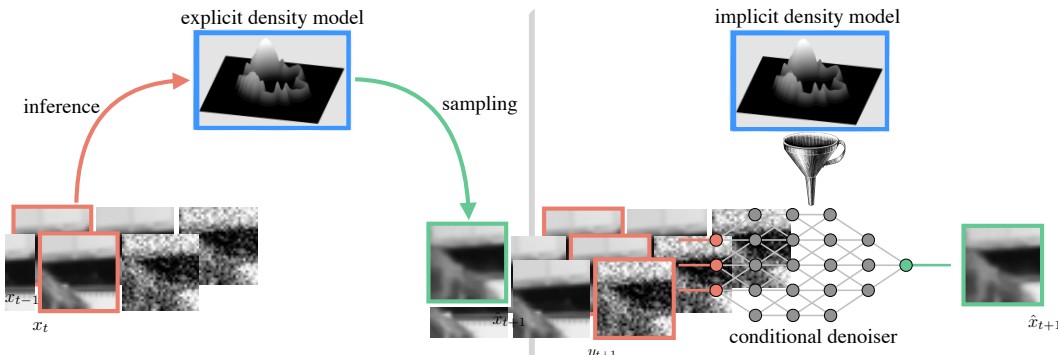

Figure 1: **Modeling frameworks for probabilistic prediction.** Both the classical and the proposed approach are trained unsupervised on image sequences and learn the distribution of the next frame conditioned on the recent past. Both approaches make predictions by sampling from this learned conditional density, but the density can be either explicit or implicit. **Left.** Traditional framework: the model is explicit and its parameters are learned through repeated iterations of inference and sampling. The learning objective is applied end-to-end: from past frames, to inferred latent representation, to generated next frame. **Right**. Proposed score-based framework: train a denoiser via regression, *i.e.*, a mapping from past frames and noisy observation to an estimate of the clean next frame. The trained denoiser approximates the score function and implicitly represents the probabilistic model (denoted by a funnel). To sample from this implicit model, iterative partial denoising gradually transforms noise into a predicted next frame.

the advent of "diffusion models", which have offered a new and highly successful paradigm for learning generative models (Song & Ermon, 2019; Ho et al., 2020; Song et al., 2021b). They have also been applied to video generation where they achieve impressive empirical results (Ho et al., 2022; Blattmann et al., 2023; Bar-Tal et al., 2024; Brooks et al., 2024). These results typically rely on huge datasets, opaque networks, and sophisticated text conditioning. In this paper, we will consider a simplified diffusion-like framework where the learned representation can be analyzed.

Diffusion models combine two important ideas: score-matching estimation and reversing a diffusion process. Score-matching bypasses the normalization constant that plagues maximum likelihood estimation (Hyvärinen, 2005). This simplification arises from considering the gradient of the logarithm of the data distribution—known as the score function—which does not depend on the problematic normalization constant. Score-based estimation has its roots in the "empirical Bayes" subfield of statistical inference, and the central role of the score of the noisy data distribution can be generalized to many other types of estimation (Raphan & Simoncelli, 2011). In diffusion models, this empirical Bayes idea is combined with a diffusion process (Sohl-Dickstein et al., 2015). Gradually adding noise to the data is equivalent to blurring the data distribution, which has a regularizing effect: it connects all the modes of the distribution and enables gradient-based sampling. Diffusion models are appealing because of their simplicity and generality: given data and a diffusion process, the scores of the noisy data can be estimated via least-squares regression, and new samples from the data distribution can be generated via stochastic score ascent. In such score-based models, the density remains implicit and is only revealed by sampling. In contrast, traditional statistical machine learning frameworks aim to build an explicit model of the data distribution. This contrast is illustrated in the case of video prediction in Figure 1. In this paper, we formulate a simple score-based approach to video prediction that facilitates visualization of the learned representation, casting some light on the adaptivity of trained networks and their ability to exploit spatio-temporal structure in image sequences.

This work studies the learned representation of a probabilistic video prediction framework, making three contributions: in Section 2, we describe a simple score-based framework for conditional density estimation applied to video prediction; then in Section 3.1, we analyze a representative case of bifurcating trajectory that arises due to occlusion boundaries; finally in Section 3.2, we reveal the adaptivity of representations in trained networks, making their performance partially interpretable.

## 2 SCORE-BASED ESTIMATION AND SAMPLING OF CONDITIONAL DENSITIES

Given access to a video dataset, *i.e.*, image sequences $\{x_t \in \mathbb{R}^d, t = 1, \ldots, T\}$, we want to predict future frames on new videos, *i.e.*, generate probable next-frames on test image sequences. To achieve this, we describe a scalable framework for learning to sample from the distribution of the next frame conditioned on the recent past. First, we assume that the temporal dependencies are local and define a corresponding network architecture. Then, we introduce a useful intermediate object, the score of the noisy observation distribution, and propose a regression objective for training a network to approximate the score. Finally, we express an iterative procedure that utilizes the trained network for sampling probable next-frames. These methods extend previous work on unconditional density modeling (Kadkhodaie & Simoncelli, 2021) to the conditional setting of video prediction. An intuitive example with low-dimensional visualizations is given in Appendix B.

### 2.1 CONDITIONING WITH A SEQUENCE-TO-IMAGE NETWORK ARCHITECTURE

We assume that temporal dependencies are local and well approximated by the Markov condition:

$$p(x_{t+1}|x_{\leq t}) = p(x_{t+1}|c_t), \tag{1}$$

where $x_{\leq t}$ denotes all the frames until time t, and $c_t = [x_t, \ldots, x_{t-\tau+1}]$ denotes the past $\tau$ frames. We enforce this conditional independence assumption by considering networks with a finite memory length. Specifically, we use deep convolutional networks that map sequences of images to estimates of the next frame. The length of the temporal dependencies captured by this sequence-to-image network can be varied by modifying the number of input frames. More details on the architecture are provided in Appendix C. In the remainder of the text, we drop time subscripts for clarity and write $x$ for the target next frame, and $c$ for the corresponding previous frames. We refer to the conditional distribution of the next frame given past conditioning, $p(x|c)$, as the data distribution.

### 2.2 LEARNING SCORE FUNCTIONS VIA DENOISING

We break down the problem of estimating the conditional distribution of the next frame into a family of easier sub-problems: given clean past conditioning frames, remove undesired noise from an observation of the next frame. Training a network on this simple resilience-to-noise objective function across noise levels yields an adaptive estimator of the score of the noisy observation distribution.

Specifically, we sample noisy observations of the next frame conditioned on the previous frames: $y|c \sim p_\sigma(y|c)$, where the noise is additive, white, and Gaussian, *i.e.*, $y = x + \sigma z$, with $z \sim \mathcal{N}(0, \mathrm{I})$. We refer to the distribution of the noisy signal given conditioning, $p_\sigma(y|c)$, as the observation distribution. It is well-known in statistical estimation that the Minimum Mean Squared Error (MMSE) denoising function is given by the posterior expectation:

$$\arg \min_{\hat{x}} \mathbb{E}\Big[||x - \hat{x}(y, c)||^2\Big] = \mathbb{E}[x|y, c]. \tag{2}$$

Remarkably, this posterior expectation can be expressed as a step in the direction of the score of the observation distribution scaled by the variance of the noise, an identity attributed to Tweedie or Miyasawa (Robbins, 1956; Miyasawa et al., 1961):

$$\mathbb{E}[x|y, c] = y + \sigma^2 \nabla_y \log p_\sigma(y|c). \tag{3}$$

This important identity links MMSE denoising with the score function, *i.e.*, the gradient of the logarithm of the observation distribution (a detailed derivation is provided in Appendix A). The key idea of the score-based framework is to exploit this link in the reverse direction: first train a denoiser to minimize MSE, and then treat the denoising residual as an approximation of the scaled score function. In Section 2.3, we will describe how to use this approximate score function to ascend the probability gradient and draw samples from the data distribution.

We consider networks that jointly process sequences of $\tau + 1$ frames and produce estimates, $\hat{x}$, of the target frame, $x$. The input sequences contain a noisy observation of the target frame, $y$, and $\tau$ past conditioning frames, $c$. Optimizing such a network on the video next-frame denoising objective, eq. (2), yields a denoising residual, $f(y, c) = \hat{x}(y, c) - y$, that approximates the scaled score functions, $\sigma^2 \nabla_y \log p_\sigma(y|c)$, and therefore depends on the noise level, $\sigma$. In the remainder

of this paper, we will integrate across noise levels. Specifically, we train networks to remove noise of arbitrary magnitude, *i.e.*, the expectation in eq. (2) is taken over the signal $x$, the noise $z$, and the noise level $\sigma$. Such networks must (at least implicitly) infer the distortion level, $\sigma$, and are called blind universal denoisers (Mohan et al., 2020). As a result, a trained universal blind denoiser contains information about a whole family of scores: $\{\nabla_y \log p_\sigma(y|c)\}_\sigma$. Recall that adding noise to the target next frame is equivalent to smoothing the data distribution with a Gaussian kernel, the more noise is added, the larger the extent of the smoothing. This family therefore contains scores at all levels of smoothness, which corresponds to a scale-space representation (Witkin, 1983) of the score of the data distribution, $\nabla_x \log p(x|c)$. In summary, we train universal blind denoisers to adaptively process signals across distortion levels and estimate a family of score functions—a property that will be exploited in the sampling algorithm described next.

### 2.3 GENERATING PREDICTED FRAMES VIA SCORE ASCENT

Assuming that we have trained a denoiser, we now turn to the question of sampling from the distribution implicit in the network, *i.e.*, predicting probable next frames conditioned on past frames. In the previous section, we showed that the residual of the optimal denoising function is proportional to the gradient of the logarithm of the observation distribution, eq. (3). We now describe an iterative procedure that starts from an arbitrary initialization and uses the denoiser to climb the gradient of the logarithm of the observation distribution towards more probable images.

Specifically, we define a conditional sampling algorithm that takes partial iterative denoising steps until reaching a sample from the data distribution, $p(x|c)$. Starting from an arbitrary image, $y_0$, this algorithm moves the candidate frame, $y_k$, in the direction of the denoising residual, and optionally adds a small amount of fresh noise, $z_k \sim \mathcal{N}(0, \mathrm{I})$. This algorithm iterates the map:

$$y_k = y_{k-1} + \alpha_k f(y_{k-1}, c) + \gamma_k z_k, \tag{4}$$

where the parameter $\alpha_k$ controls the step-size and the parameter $\gamma_k$ controls the amplitude of the additive noise. Importantly, we choose $\gamma_k$ so as to reduce the effective noise level on each iteration. Adding noise along the sampling iterations avoids getting stuck in local maxima and promotes exploration. A more detailed description of these sampling parameters is provided in Appendix E.

This procedure follows the estimated scores given by a trained denoiser and gradually reduces the effective noise level of a candidate frame until it reaches a sample of the next frame. As this algorithm progresses, the candidate frame effectively traverses the scale-space family of observation distributions from large to small noise level. Such a coarse-to-fine iterative refinement approach enables local search in complex high-dimensional landscapes. Importantly, this annealing schedule is set by the network itself: at each iteration, the step size depends on the magnitude of the denoising residual, this magnitude is set by the blind universal denoiser, *i.e.*, it automatically adapts to the effective noise level of the candidate frame.

## 3 VISUAL TEMPORAL PREDICTION UNDER UNCERTAINTY

### 3.1 HANDLING OCCLUSIONS ON A PROCEDURAL DATASET

Occlusion boundaries are an inevitable result of image formation and they provide strong cues which are exploited by biological visual systems to infer the depth ordering and relative motion of objects in a scene. Traditional video prediction methods are based on a local translation model (*e.g.*, block matching methods used in MPEG coders, or optic flow methods of computer vision), and have difficulties at occlusion boundaries. In fact, occlusions are a major source of errors for methods that can not represent motion discontinuities and do not account for ambiguous occlusion relationships.

**Moving leaves.** We consider a reduced scenario focused on the challenge of ambiguous occlusions. Our aim is to demonstrate that, unlike previous methods, the score-based inference framework can handle occlusions and that it makes decisions on ambiguous sequences. Building on the "dead leaves" model of the natural scene statistics literature (Matheron, 1975; Lee et al., 2001), we design a dataset of moving disks. Each image sequence is composed of two moving disks whose trajectories may intersect, resulting in one occluding the other (see Figure 2). Disks are randomly placed on an image canvas, a random depth is assigned to each disk. Each disk is then moved along a randomly

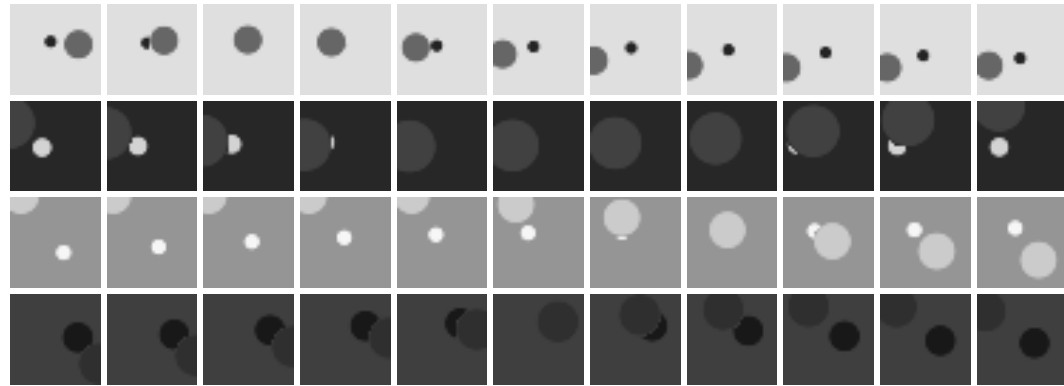

Figure 2: **Moving leaves dataset.** Example image sequences from our synthetic dataset. Each sequence contains two disks moving along smooth curves and occluding each other. The disks move against a blank background and the larger disk always occludes the smaller one.

chosen smooth trajectory to generate image sequences that respect occlusion relationships. Trajectories are sampled from a Gaussian process, and their average speed scales inversely with depth (*i.e.*, objects at a further distance moving more slowly). For simplicity, each disk is assumed to move within a plane at its pre-assigned depth, and thus its size in the image does not change. These image sequences contain two reliable, albeit indirect, cues to depth: the projected disk size, and the speed of disk motion. The luminance of each disk and of the background are selected randomly, from a uniform distribution. More details on this dataset are provided in Appendix D.

**Deciding on occlusion boundaries.** A U-net with two conditioning frames ($\tau = 2$) was trained on the moving leaves dataset. We generate samples of probable next frame using the sampling algorithm described in Section 2.3. A detailed description of the training and sampling algorithms is available in Appendix E. We evaluate the network on new moving disk sequences designed to probe the network's ability to handle ambiguous trajectories that bifurcate into two probable future sequences. These probe sequences contain two clean conditioning frames with disks that are moving towards each other on a collision course. We control the relative size of the two disks and put pure noise in place of the target next frame. As expected, when the depth ordering is unambiguous, the network correctly estimates the most probable next frame (see Appendix F Figure 11).

A more challenging and interesting situation arises when the depth ordering is ambiguous. Example samples around such an ambiguous occlusion boundary are displayed in Figure 3. In this example there are no cue as to which disk will occlude the other. As expected, the estimated next frame obtained via one-step denoising is a blurry mixture between the two possibilities. This corresponds to the least squares optimal solution, eq. (2), which averages over the posterior. In contrast, generated next frames obtained via iterative sampling contain sharp occlusion boundaries, with either disk occluding the other at about equal frequency. The network effectively decides on occlusions and generates diverse samples that, unlike one-step denoising, do not compromise between the two possibilities. Moreover, the network has learned from the training dataset that smaller disks are likely to be occluded and respects this property in sampled sequences. This highlights the power of the score-based estimation and sampling framework compared to deterministic one step prediction.

**Selecting more likely options with higher frequency.** We vary the degree of ambiguity of a disk occlusion by controlling the relative size of two disks moving towards one another. The results are summarized in Figure 4 and show that the network adapts its predictions to the relative size difference between the two disks. When the measurement are unambiguous, the samples are almost deterministic, and on more ambiguous measurements, the samples are gradually more stochastic.

**Recursive generation of sequences.** Longer sequences can be generated by recursively applying the network to its own predicted frames, and these sequences can be evaluated for their temporal coherence. As illustrated in Figure 5, recursive applications of the one-step denoiser produce blurry

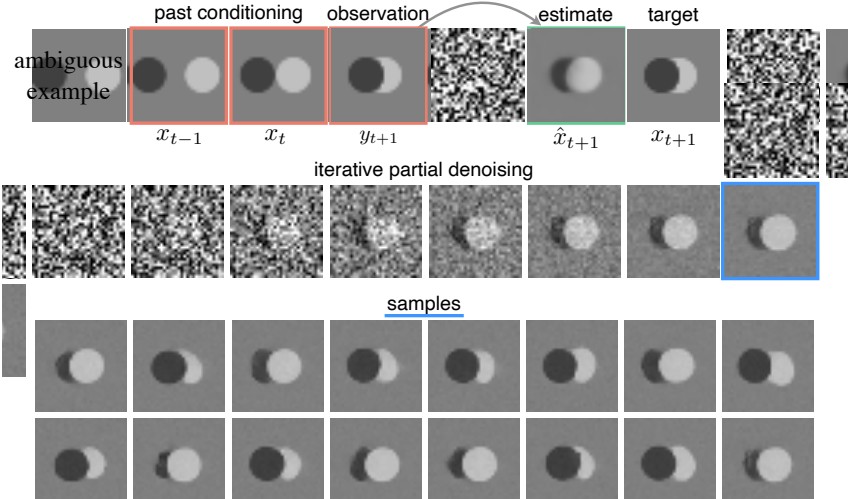

Figure 3: **Samples around ambiguous occlusion boundary.** Predicting an ambiguous sequence of moving disks. **Top.** Two conditioning frames contain disks of equal size moving towards each other. The network takes as input the conditioning frames and pure noise in place of next frame (highlighted in red). The network outputs an estimated next frame (highlighted in green). **Middle.** Intermediate steps of the iterative denoising procedure and corresponding sampled next-frame (highlighted in blue). The score-based sampling algorithm uses the same conditional denoiser network as above but takes partial denoising steps. **Bottom.** Example samples of probable next-frame generated via iterative partial denoising starting from different random initializations.

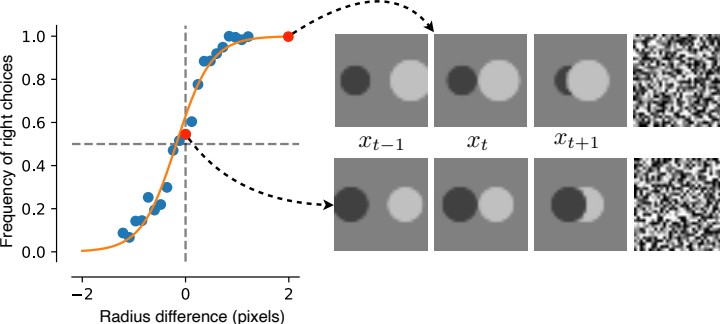

Figure 4: **Decisions preceding occlusion events.** Frequency of right disk occlusion in next-frame samples as a function of the difference in radius between the two disks (averaged over 64 samples). Highlighted red points correspond to the unambiguous examples from Figure 11 and the ambiguous example from Figure 3. The orange curve is a logistic function fit to the choice data, it describes the network's sensitivity to the disk's relative size difference.

estimates that collapse to an uninformative image after a few recursive prediction steps. In contrast, recursive samples obtained by sampling (iterative partial denoising) produce a coherent image sequence with disk motion and occlusions (not shown in this particular example). However, the network is limited by its short memory length, here only accessing two past conditioning frames. As a result samples from the network tend to drop or modify disks that are occluded in the observed past conditioning frames. As illustrated, disks tend to reappear distorted when emerging from occlusion. Capturing longer-term temporal dependencies requires networks with longer memories. Indeed, two conditioning frames are insufficient to capture acceleration.

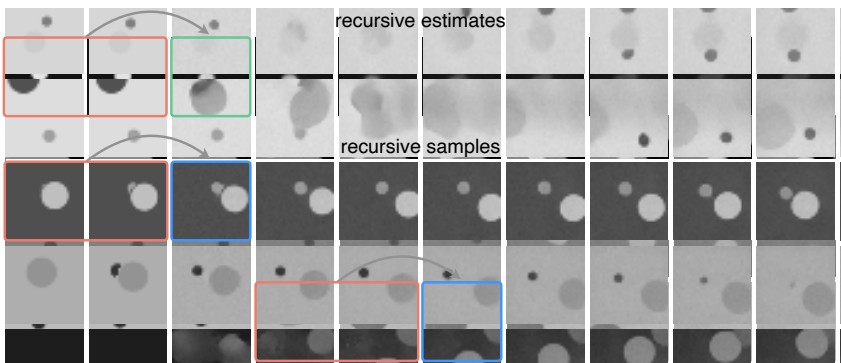

Figure 5: **Recursive generation of moving disk sequences.** Coherent sequences can be generated via recursive sampling, but not via recursive estimation. In all three examples the first two frames come from the test dataset and the successive frames are generated recursively, using the previous two frames as conditioning. **Top.** Frames are generated by direct application of the denoiser. For the first step of this recursive process, the inputs to the conditional denoiser are highlighted in red (the noise observation, $y_{t+1}$, is not shown) and the estimated next frame is highlighted in green. **Middle.** Frames are generated via iterative partial denoising. The generated next frame (highlighted in blue) is a sample from the conditional distribution of probable next frame. **Bottom.** Another example of recursive sampling. The fourth step in the recursive sampling process is highlighted and shows that after full occlusion of the small disk, its size, color and location are often altered.

## 3.2 ADAPTIVE PROBABILISTIC PREDICTION OF NATURAL VIDEOS

**Conditional denoising performance.** Several U-nets with varying conditioning lengths were trained for next frame prediction on a generic dataset of small natural image sequences (see description in Appendix D). Estimated next frames for denoisers with varying number of conditioning frames are shown for an example image sequences in Figure 6. This result holds in general over the whole test set, as evidenced by the summarized performance plot of these networks. We quantify performance according to Peak Signal to Noise Ratio[2]. In the unconditional case (*i.e.*, $\tau = 0$), the network only exploits static image structure and performance is linear in terms of input-output PSNR, with slope of approximately one-half, and meets the identity line around 40 dB. For conditional image denoising, there is a gradual performance improvement that is particularly salient at low input PSNR values. The largest increases in output PSNR is obtained from conditioning on one baseline image (*i.e.*, $\tau = 1$). Adding a second conditioning frame (*i.e.*, $\tau = 2$) improves performance further, which is indirect evidence that the network is making use of motion information. The benefits of conditioning reach a plateau with three past frames. Notice that trained denoiser networks with two and three conditioning frames match the performance of networks trained just for prediction (*i.e.*, two input frames $x_{t-1}, x_t$, no noisy observation $y_{t+1}$), indicated by a horizontal dashed line at 28dB. Remarkably, at very low input PSNR values the network automatically revert to pure temporal prediction, ignoring the uninformative observation.

**Spatio-temporal adaptive linear filtering.** Visual motion carries important information for video prediction and networks should learn to utilize this information. We analyze the action of a trained network on an example image sequence to reveal that it is exploiting visual motion. Our U-nets are bias-free and effectively apply an adaptive linear filter to their input (Mohan et al., 2020). Specifically, the conditional denoiser can be expressed as an input dependent linear function. For convenience, we split this linear function into two parts, one applied to the conditioning frames (with $c$ subscript) and one applied to the noisy observation (with $y$ subscript):

$$\hat{x}(y,c) = \hat{x}_y + \hat{x}_c, \text{where } \hat{x}_y = \nabla_y \hat{x}(y,c) \cdot y, \text{ and } \hat{x}_c = \nabla_c \hat{x}(y,c) \cdot c. \tag{5}$$

Those input-output Jacobian matrices can be thought of as effective linear filters adapted to the input. Evaluating those Jacobians reveals that filters that are oriented in space-time and track the motion

---

[2]PSNR is a standard quality metric measured in decibels (dB) which expresses the logarithm of the mean squared error (MSE) relative to the range of the signal: $\mathrm{PSNR}(x, \hat{x}) = 10 \log_{10}(\mathrm{I}_{\mathrm{range}}^2/\mathrm{MSE}(x, \hat{x}))$, where $\mathrm{I}_{\mathrm{range}}$ is the range of possible pixel values of the image.

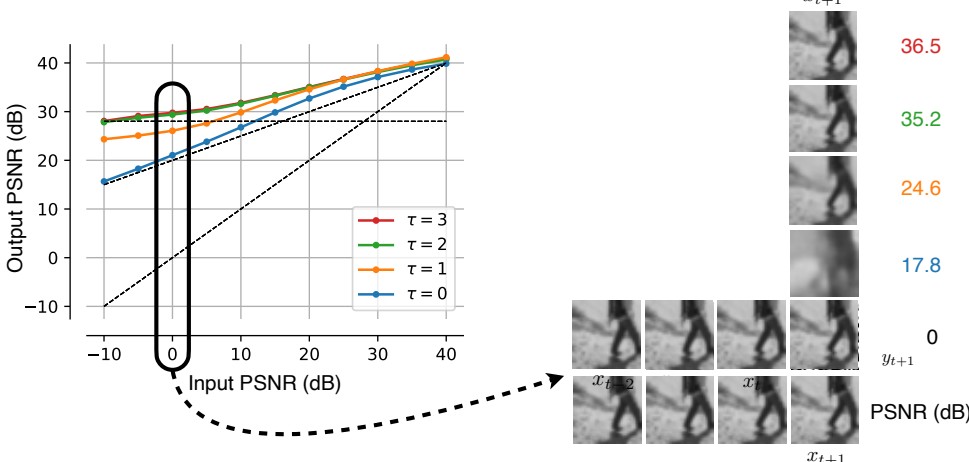

Figure 6: **Peformance of conditional denoiser, for varying numbers of conditioning frames, $\tau$.**
**Left.** Input-output PSNR curves summarize the test performance of trained denoisers. The horizontal axis represents noise level (lower PSNR value correspond to stronger distortion). The vertical axis represents performance (higher PSNR value corresponding to better denoising). Dashed black lines indicate the identity, a slope of one half (expected for single image denoiser performance), and a horizontal line at 28dB corresponds to the performance of the same network trained for prediction alone. **Right.** Horizontally: example image sequence, noisy observation and target next frame at 0dB input PSNR. Vertically: estimated next frame for denoisers with varying number of conditioning frames (PSNR indicated with color codes matched to graph on left). Longer memory results in higher quality denoising, showing that the denoiser can exploit spatio-temporal image structure.

of the pattern in image sequences, which is analogous to the observations made on video denoising networks (Sheth et al., 2021). One row such example adaptive filter is displayed in Figure 7 and shows evidence of a visual motion computation.

**Adaptively weighing evidence by reliability.** The network utilizes both past conditioning frame, $c$, and noisy observation, $y$, and we now quantify how these two sources of information are combined to produce an estimated next frame. The posterior over these two cues can be decomposed as:

$$p(x|y,c) = \frac{p(x,y,c)}{p(y,c)} = \frac{p(y|x)p(x,c)}{p(y,c)} = \frac{p(x|y)p(x|c)}{p(x)} \frac{p(y)p(c)}{p(y,c)}, \tag{6}$$

where the second step uses the conditional independence of $y$ and $c$ given $x$, $p(y|x,c) = p(y|x)$. Notice that the posterior over $y$ and the posterior over $c$ are combined multiplicatively, and that they are modulated by an interaction term that quantifies how independent these two cues are. The corresponding decomposition in the case of squared denoising error is:

$$||x - \hat{x}(y,c)||^2 = ||x - \hat{x}_c||^2 + ||x - \hat{x}_y||^2 - ||x||^2 + 2\langle \hat{x}_y, \hat{x}_c \rangle, \tag{7}$$

where we reuse the notation from eq. (5). This expression is a partition of variance and it reveals how each cue contributes to the overall denoising performance. We evaluated the network performance on three probe sequences at different noise levels. We also computed a local linear approximation of the network at each of these levels and computed the first two terms on the right hand side of equation 7. The results are displayed in Figure 8 and show that the network appropriately combines evidence by weighing each cue by its reliability. These denoising curves show that conditioning has more impact on the estimates at high noise level. As the noise level is reduced, the model relies gradually more on the observation and finally ignores the conditioning altogether. Importantly, this dependency is modulated by the predictibility of the image sequence. In summary, the network performs blind evidence integration: it automatically adapts to both the distortion level in the noisy observation, and to the amount of predictive information in the past conditioning frames.

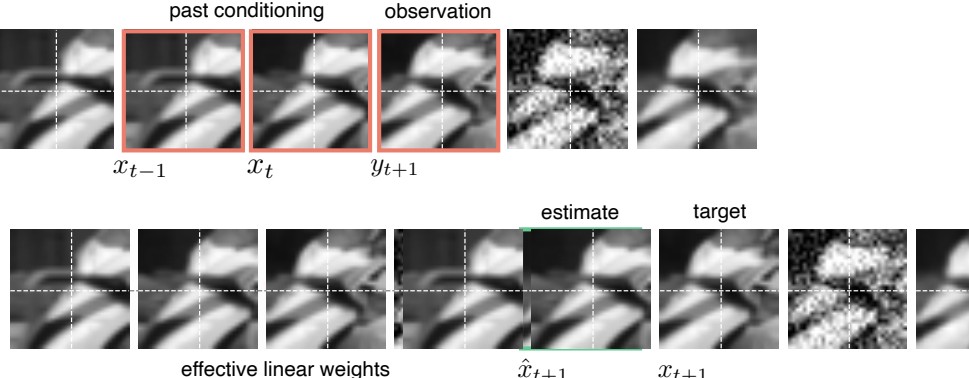

Figure 7: **Spatio-temporal adaptive filtering.** Visualization of the effective linear weights used to compute the central denoised pixel of an example image sequence. **Top.** Image sequence from the test set, a pattern is moving to the left. Dashed white lines highlight the central pixel of each frame for reference. The two past conditioning frames are clean and the noisy observed frame has a PSNR of about 16 dB. **Bottom.** Effective linear filter used by the network to weight the conditioning frames and the noisy observation, $y_{t+1}$, to estimate the center pixel of the next frame $\hat{x}_{t+1}$. Each pixel of the estimate is effectively computed as a weighted sum of input pixels (only the weighting corresponding to the center pixel is shown here). Notice that the effective filter locally averages the noisy observation and focuses on the part of the previous image displaced to the right (corresponding to leftward motion of the pattern). The estimated frame has a much higher PSNR of about 25.5 dB.

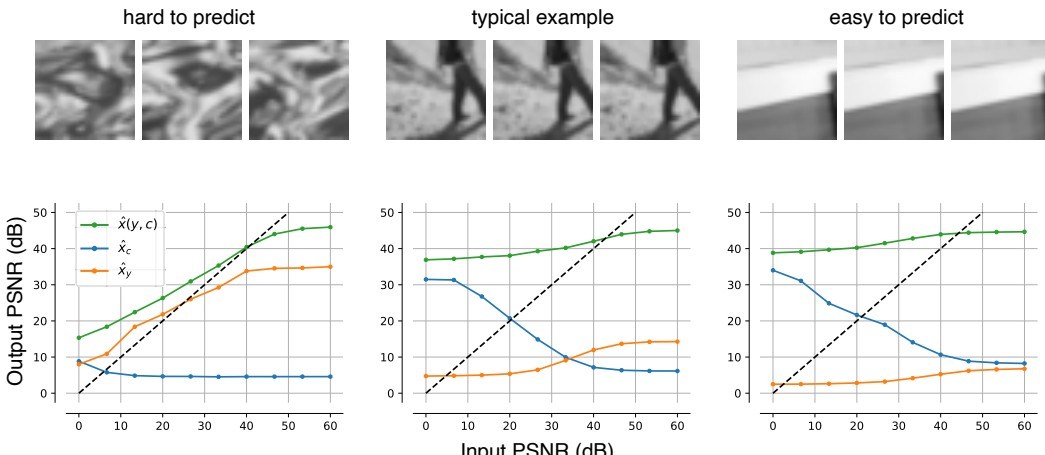

Figure 8: **Adaptive cue combination.** Local linear analysis of information integration from two different sources, past conditioning and noisy observation. In green the denoising performance for a single probe image sequence as in Figure 6. In orange the denoising performance due to noisy observation only, which increases with input PSNR. In blue the denoising performance due to past conditioning frames only, which decreases with input PSNR. These two curves are calculated using a local linear approximation as described in eq. (5). The level at which the orange and blue curve cross depends on how difficult the prediction is, demonstrating that the network adaptively weights evidence by reliability. **Top.** A texture image sequence with non-rigid complex motion. The orange and blue curves cross around 0dB input PSNR. **Middle.** A person walking to the right on a stable background. The orange and blue curves cross around 30dB input PSNR. **Bottom.** A static pattern consisting of three smooth regions separated with straight boundaries. The orange and blue curves cross around 60dB input PSNR.

## 4 RELATED WORK

**Video prediction.** Estimating optic flow is an essential component of standard video codecs and deep network solutions have been proposed (Wu et al., 2022). But explicit motion warping fails around occlusion boundaries and non-rigid objects (Baker et al., 2011). Direct next-frame prediction has been explored with generative adversarial networks (Mathieu et al., 2016). But the saddle point objective function is difficult to train and suffers from mode collapse (Goodfellow, 2016). Methods based on variational autoencoders (Kingma & Welling, 2014) tend to suffer from posterior collapse (Wang et al., 2021). Normalizing flows (Dinh et al., 2014) are limited to architectures that admit tractable Jacobians at each layer (Papamakarios et al., 2021).

**Diffusion based video generation.** Recent work on video generation uses diffusion models (Harvey et al., 2022; Höppe et al., 2022) and has been applied to video prediction Voleti et al. (2022). These studies obtain impressive empirical results but rely on complex architectures and sampling algorithms that make their results difficult to interpret. Mathematically principled approaches to video prediction have been proposed Chen et al. (2024). Here we describe a simple score-based framework that facilitates visualization of the learned representation, casting some light on the adaptivity of trained networks and their ability to exploit spatio-temporal structure in image sequences.

## 5 DISCUSSION

We describe a simple framework for estimating and sampling from the distribution of probable next-frames in image sequences and use this framework to visualize and study the adaptivity of deep network representations. We show how deep denoising networks trained on a simple regression task—minimizing mean squared error—implicitly represent the score of the data distribution. We put this implicit density model to use by generating probable next-frames. Experiments on synthetic data show that the sampled next-frames correctly handle ambiguous situations, choosing a depth order for objects that occlude each other, which contrast with traditional methods that blur such ambiguous cases. Furthermore, we showed that denoisers display hallmarks of probabilistic computation: the network adaptively combines information from past conditioning and noisy observations, appropriately weighting them according to their reliability. The adaptability to the amount of predictive information is remarkable: the network evaluates its own ability to extract evidence from past frames. The simplicity of the score-based framework suggests that density estimation and sampling—although very challenging in high-dimensions—may be tackled with elementary tools: (non-linear) least squares regression and (coarse-to-fine) gradient ascent. This is only possible because i) noise offers a local learning signal and enables sampling, and ii) the network architecture has inductive biases that are appropriate for natural image sequences.

**What is noise good for?** The functional role played by noise plays in both estimation and sampling and can be summarized by three properties: regularity, diversity, and locality. First, adding noise smooths the energy landscape, making it easier to learn and to climb with local search methods (both estimation and sampling are gradient based). Second, adding noise during sampling drives exploration, allowing the samples to escape from local maxima, and promoting diversity. Third, resilience to noise is a local objective in the sense that it brings random Gaussian samples towards the data distribution incrementally. Such locality in the space of densities offers a powerful simplification: the transformation from data to Gaussian noise and back to data is broken down into small steps. Denoising provides a learning signal for each of these intermediate steps and allows learning to proceed in parallel. We conjecture that this locality of denoising underlies the statistical efficiency of score estimation compared to traditional approaches based on end-to-end objective functions.

**Implicit bias of the architecture.** The performance achieved by deep networks on (conditional) image denoising is remarkable. Accurate denoisers provide good approximations of score functions and fuel the success of diffusion models. Understanding how the architecture of deep denoising networks enables such denoising performance is of great interest. Local linear analysis affords partial access to the adaptive network representation, but it does not elucidate the formation of these representations. Further study of deep network implicit biases is necessary, and the simple framework and visualization methods described here offer a starting point. Finally, it would be interesting to leverage video prediction for extracting abstract representations. Considering an encoder-decoder architecture instead of a direct sequence-to-image mapping is a promising direction.

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

## A    DERIVATION

In this section, we provide a detailed derivation of the conditional Tweedie/Miyasawa relationship, equation (3). We will use the fact that for any function $h$:

$$\nabla_y h(y) = h(y) \nabla_y \log h(y). \tag{8}$$

First, express the observation density as a convolution of the data density with the measurement density:

$$p_\sigma(y|c) = \int p(x|c) p_\sigma(y|x, c) dx, \tag{9}$$

differentiate on both sides with respect to the noisy variable, $y$, and then use eq. (8):

$$\nabla_y p_\sigma(y|c) = \int p(x|c) \nabla_y p_\sigma(y|x, c) dx \tag{10}$$

$$= \int p(x|c) p_\sigma(y|x, c) \nabla_y \log p_\sigma(y|x, c) dx, \tag{11}$$

divide on both sides by the observation density, $p_\sigma(y|c)$, and use Bayes rule:

$$\frac{\nabla_y p_\sigma(y|c)}{p_\sigma(y|c)} = \int \frac{p(x|c) p_\sigma(y|x, c)}{p_\sigma(y|c)} \nabla_y \log p_\sigma(y|x, c) dx \tag{12}$$

$$= \int p(x|y, c) \nabla_y \log p_\sigma(y|x, c) dx, \tag{13}$$

use eq. (8) again, and write the integral as an expectation:

$$\nabla_y \log p_\sigma(y|c) = \mathbb{E}_x[\nabla_y \log p_\sigma(y|x, c)|y, c]. \tag{14}$$

In the case of a Gaussian measurement density, and using the conditional independence of $y$ and $c$ given $x$, we have:

$$p_\sigma(y|x, c) = p_\sigma(y|x) = g_\sigma(y - x) = \exp(-\frac{(y - x)^2}{2\sigma^2})/\sqrt{2\pi\sigma^2}, \tag{15}$$

take the logarithm and the gradient:

$$\nabla_y \log p_\sigma(y|x, c) = (x - y)/\sigma^2, \tag{16}$$

plug this into eq. (14):

$$\nabla_y \log p_\sigma(y|c) = (\mathbb{E}[x|y, c] - y)/\sigma^2, \tag{17}$$

which proves eq. (3).

## B    INTUITIVE EXAMPLE IN ONE DIMENSION

In this section, we provide an intuitive one dimensional illustration of the score-based estimation and sampling framework described in Section 2. For these examples we consider the simpler density estimation problem, *i.e.*, without conditioning.

**Estimation.**    A simple example consisting of a one dimensional distribution comprising two point masses is displayed in Figure 9. At very high noise levels the score points in the direction of the origin. At lower levels of noise, the score points towards the closest of the two point masses of the bimodal distribution. The transition between these two regimes is continuous and smooth. We consider universal blind denoisers, *i.e.*, mappings that automatically adjust to the noise level. The optimal denoiser, $\hat{x}(y)$, should integrate over noise levels:

$$\hat{x}(y) = y + \int \sigma^2 \nabla_y \log p(y|\sigma) p(\sigma|y) d\sigma. \tag{18}$$

Let us numerically evaluate this optimal denoiser and specify a non-informative prior for the noise level: $p(\sigma) \propto 1/\sigma$, *i.e.*, a log-uniform distribution. In practice, when considering image denoising, estimating the noise level is a simple one dimensional problem that is well constrained by the many observed pixels. We therefore approximate this optimal denoiser and use a maximum a posteriori (MAP) plug-in estimator for the effective noise level: $\hat{\sigma} = \operatorname{argmax}_\sigma p(\sigma|y)$.

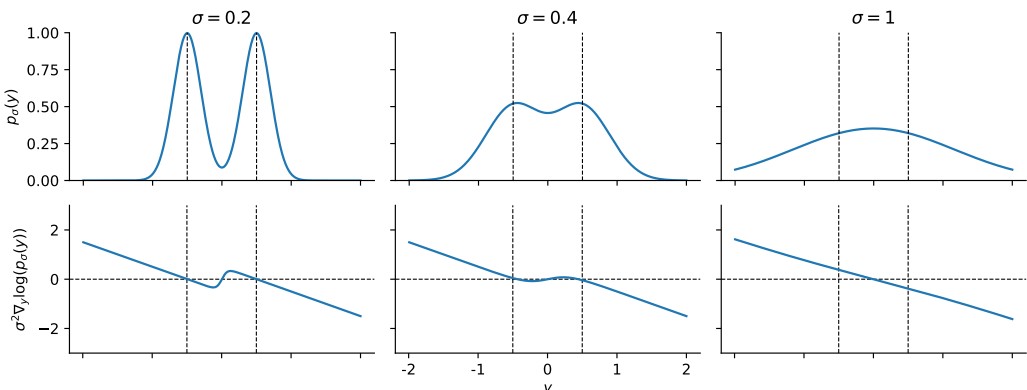

Figure 9: **Bimodal distribution and score across noise level.** One dimensional illustration of the approximation problem. **Top.** Bimodal distribution consisting of two point masses placed at $-1/2$ and $1/2$ (vertical dashed lines). From left to right, the distribution is convolved with a Gaussian kernel of increasing width, which corresponds to adding independent Gaussian noise of increasing standard deviation. **Bottom.** Optimal denoising step at corresponding noise levels, *i.e.*, the score scaled by the noise variance. The score is the gradient of the log probability with respect to the variable, $\nabla_y \log p_\sigma(y)$. The sampling process described in Section 2.3 uses these scores to gradually reduce noise (*i.e.*, traverse the family of noisy distributions from right to left on the top row).

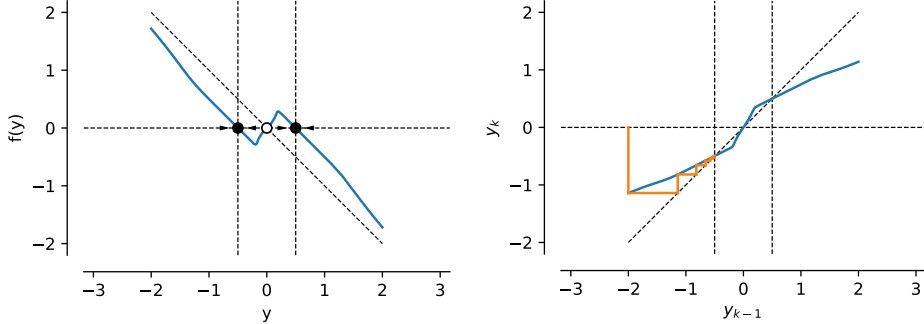

Figure 10: **Sampling bimodal distribution via iterative partial denoising Left.** Denoising residual of the blind universal denoiser for the bimodal distribution displayed in Figure 9. The denoiser was approximated with MAP plug-in estimator for the standard deviation of the noise (see text). There are two stable fixed points on the data point masses of the target distribution and an unstable fixed point between the two. **Right.** One dimensional illustration of the sampling procedure that iterates the map: $y_k = y_{k-1} + 0.5 f(y_{k-1})$. An example sequence $y_{k k}$ starting at $y_0 = -2$ is illustrated in orange. The sampling algorithm reaches one of the fixed points depending on initial conditions.

**Sampling.** This one dimensional bimodal distribution also enables visualization of the sampling algorithm. The algorithm proceeds by iterative partial denoising and is detailed in Section E. Using the universal blind denoiser described in the estimation section, we can define an denoising residual, $f(y) = y - \hat{x}(y)$, that approximates an adaptive score function:

$$f(y) \approx \hat{\sigma}(y)^2 \nabla_y \log p(y|\hat{\sigma}(y)). \tag{19}$$

For simplicity, we use a fixed step size, $\alpha = 0.5$, and no additional noise, $\beta = 1$. With these choices, the sampling algorithm iterates the map: $y_k = y_{k-1} + 0.5 f(y_{k-1})$. This map choose between the two point masses depending on the initial condition, as illustrated in Figure 10.

## C   DESCRIPTION OF ARCHITECTURES

For all numerical experiments, we use U-nets composed of convolutional layers and rectifying non-linearities at three spatial scales (Ronneberger et al., 2015). We consider sequence-to-image networks, *i.e.*, mappings from $\tau$ past frames and one noisy observation to an estimated next frame: $\hat{x}_\theta : \mathbb{R}^{d \times \tau} \times \mathbb{R}^d \to \mathbb{R}^d$.

**Conditioning.**   The networks take as input the past conditioning frames concatenated with the noisy observation, *i.e.*, five dimensional tensors of shape: [batch, channel, time, height, width], where time $= \tau + 1$. The size of the first layer spatio-temporal convolution kernel is adjusted to match the number of input frames: $(\tau + 1) \times 3 \times 3$. Concatenating past frames as additional input is a simple approach to conditioning and it has the advantage of being well suited to temporal prediction: conditioning frames have the same shape as the noisy observation, and local convolutional processing can make use of these two sources of evidence for the denoising task.

**Details.**   At each scale, the networks cascade two blocks, each containing a layer of spatial convolution, then layer-normalization, and finally rectification. Specifically, the convolutional filters are of size $3 \times 3$ in space. The first stage contains 64 filters. Each successive stage of the encoder doubles the number of channels and downsamples the image by a factor 2 in each spatial dimension. Each stage of the decoder reverses this, upsampling spatially, and halving the number of channels. Each stage of the decoder also combines information from the upsampled coarse stage with the output of the encoding at the corresponding stage. The network directly outputs an estimated next frame, *i.e.*, there is no input-output skip connection. In total these networks contain almost 2 million parameters.

**Homogeneity.**   Importantly, we remove all bias terms from the networks, *i.e.*, there are no additive constants in the convolutional layers or in the normalization layers. Such bias-free networks are positively homogeneous of order 1, *i.e.*, scaling the input by a positive number also scales the output by the same amount. This simplification of the network architecture was observed (Mohan et al., 2020) to facilitate generalization across noise levels, moreover, it enables the local linear analysis presented in section 3.2.

## D   DESCRIPTION OF DATASETS

**Moving Leaves.**   Each image sequence contains two disks on a blank background that move and occlude each other. In each sequence, both disks are assumed to have the same 3D physical size but their distance to the imaging plane is randomized. As a result, each disk's projected 2D size is a reliable, although indirect, cue to its distance in the scene. The disk with larger projected 2D size always occlude the smaller one. The trajectories of each disk are sampled from Gaussian processes, and their average speed of motion scales inversely with distance. Thus, speed in the image plane provides an additional (albeit weaker) cue to a disk's distance.

This synthetic dataset contains $10^5$ image sequences, and is split into train and test set with at a 9:1 ratio. Each sequence is composed of 11 frames and is of size $32 \times 32$ pixels. For each image sequence, luminance and depth are sampled uniformly at random. The spatial positions (x and y) are sampled from a Gaussian process, and the bandwidth of the GP scales with depth (smoother slower trajectories for objects further away). In each image sequence, there is at least one frame where half of smaller disk is occluded by the larger disk.

There are several benefits to using such a procedural dataset: it enables sampling a distribution along continuous features such as size, speed, texture, *etc.*; and it is easy to change the rules and control the difficulty of the task. For example allowing the disks to move along the depth axis, thereby decoupling size from distance; or changing the occlusion rule (e.g., assuming the brighter disk always occludes the darker, or making the occlusion rule stochastic; or changing the number of disks, *etc.*). This dataset is also useful to study the representation of fast small objects, which are subject to temporal aliasing. The moving leaves procedural dataset was designed to illustrate the fact that, even in elementary mechanical scenarios, probabilistic modeling is appropriate. Indeed, measurements are always incomplete and the probabilistic framework can readily handle this ambiguity.

**DAVIS32.** We use the DAVIS dataset (Pont-Tuset et al., 2017), which was originally designed as a benchmark for video object segmentation. Image sequences in this dataset contain diverse motion of scenes and objects (eg., with fixed or moving camera, and objects moving at different speeds and directions), which make next frame prediction challenging. Small image sequences were cropped out of the original DAVIS dataset, each sequence is composed of 11 frames of size $32 \times 32$ pixels. The spatial crops were taken at 21 spatial locations and 3 different spatial scales. The frames were mapped to grayscale and their intensity rescaled to lies in $[0, 1]$. These approximately $5 \cdot 10^4$ sequences were divided into about 36 thousands training sequences (or about 400 thousand frames), and about 5 thousands test sequences (or about 55 thousand frames).

# E  DESCRIPTION OF OPTIMIZATION AND SAMPLING

**Training procedure.** The networks are trained to minimize mean squared error weighted by inverse variance of the noise. For each frame in a minibatch, the logarithm of the variance of the noise is sampled uniformly in the range $[10^{-5}, 10^2]$ (corresponding to input PSNRs in [-20dB, 50dB]). These choices of MSE weighting and of variance distribution is equivalent, up to reparameterization, to the commonly used likelihood weighting (Song et al., 2021a). It can be shown that this training procedure minimizes an upper bound on the conditional entropy of the future frame $x$ given the past $c$ (up to constants). Notice that these choices emphasize the samples with low noise by sampling them more frequently and by amplifying their contribution to the loss. In practice, small gains in denoising performance, especially in the low noise regime, can have a large impact on the quality of generated samples. The models are trained for 1000 epochs on either the moving leaves, or the DAVIS32 datasets using the Adam optimizer (Kingma & Ba, 2015) with default parameters and a learning rate $\eta = 1 \cdot 10^{-3}$. The learning rate is halved on epoch 500, and then every 100 epochs. A detailed description is presented in Algorithm 1.

---

**Algorithm 1** Estimation of the family of score functions via denoising

1:  **inputs**: $\mathcal{D} = \{x_t\}_{1 \leq t \leq T}, \sigma_{min}^2, \sigma_{max}^2, \eta$
2:  **initialization**: Set $n = 1$, and draw $\theta$
3:  **while** $n \leq N$ **do**
4:      $(x, c) \sim \mathcal{D}$                                    ▷ Draw target next frame and conditioning frames
5:      $\log \sigma^2 \sim \mathcal{U}\big[\log(\sigma_{min}^2), \log(\sigma_{max}^2)\big]$, and $z \sim \mathcal{N}(0, \mathrm{I}_d)$        ▷ Draw noise level and noise
6:      $y = x + \sigma z$                                          ▷ Generate noisy observation
7:      $\mathcal{L} = ||x - \hat{x}_\theta(y, c)||^2 / \sigma^2$              ▷ Compute the weighted MSE loss
8:      $\theta \leftarrow \mathrm{Adam}(\theta, \mathcal{L}, \eta)$                      ▷ Update the parameters
9:      $n \leftarrow n + 1$
10: **end while**
11: **return**: $\hat{x}_\theta$

---

**Sampling algorithm.** New samples are generated by stochastic score ascent, using a family of learned approximate scores. The procedure is initialized on pure noise of large variance, $\sigma_\infty = 3$, The algorithm proceeds by taking steps in the direction of the denoiser residual and gradually reducing the effective noise level. This effective noise level at iteration $k$ is defined as the root-mean-square value of the residual: $\sigma_k = ||f(y_{k-1}, c)|| / \sqrt{d}$, with $d$ the dimensionality of the target signal. The sampling algorithm terminates when the effective noise level falls below a threshold, we used $\sigma_0 = 0.01$ in our experiments. The amount of noise added at each step of the sampling procedure is controlled by an additional parameter, $\beta \in [0, 1]$, which sets the proportion of injected noise and plays the role of an inverse temperature. Specifically, at each iteration the amplitude of the additive noise is set to: $\gamma_k^2 = ((1 - \beta\alpha_k)^2 - (1 - \alpha_k)^2)\sigma_k^2$. This choice ensures that the effective noise level is reduced at each iteration, it can be thought of as an self-adapting annealing schedule (see Kadkhodaie & Simoncelli, 2021 for the derivation). We used $\beta = 0.5$ for all experiments in this paper. We use a geometric schedule for the step-size, $\alpha_k$. A detailed description is presented in Algorithm 2.

# F  ADDITIONAL EXAMPLES

---

**Algorithm 2** Sampling via iterative partial denoising (Kadkhodaie & Simoncelli, 2021)

1: **inputs**: $\hat{x}, h_0, \beta, \sigma_0, \sigma_\infty$
2: **initialization**: Set $k = 1$, and draw $y_0 \sim \mathcal{N}(0.5, \sigma_0^2 I_d)$
3: **while** $\sigma_k \geq \sigma_\infty$ **do**
4:      $\alpha_k = \frac{h_0 k}{1 + h_0(k-1)}$          ▷ Set step size with geometric schedule
5:      $s_k = \hat{x}(y_{k-1}, c) - y_{k-1}$      ▷ Compute the weighted score, *i.e.*, denoiser residual
6:      $\sigma_k^2 = ||s_k||^2/d$          ▷ Compute the effective noise variance
7:      $\gamma_k^2 = \left((1 - \beta\alpha_k)^2 - (1 - \alpha_k)^2\right)\sigma_k^2$      ▷ Set noise level st. effective noise level decreases
8:      $z_k \sim \mathcal{N}(0, I_d)$          ▷ Draw noise
9:      $y_k = y_{k-1} + \alpha_k s_k + \gamma_k z_k$      ▷ Perform a partial denoising step and add noise
10:      $k \leftarrow k + 1$
11: **end while**
12: **return**: $x_k$

---

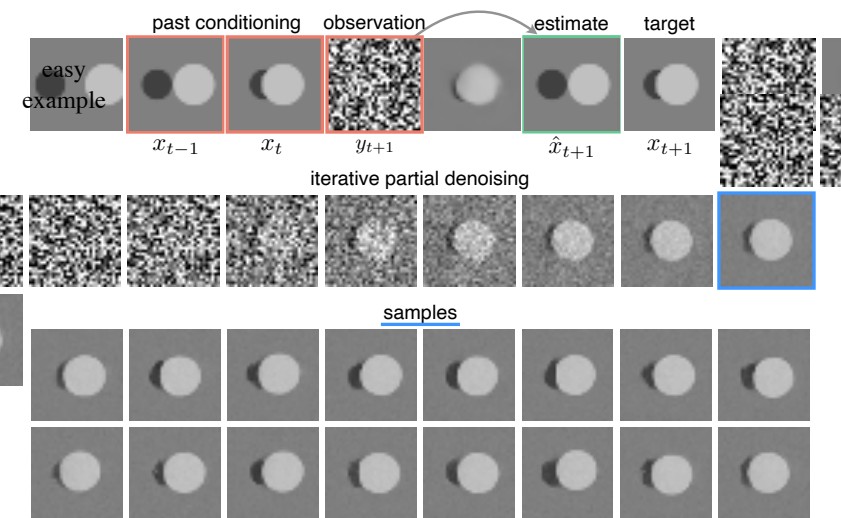

Figure 11: **Samples around unambiguous occlusion boundary.** Predicting an unambiguous disk occlusion. **Top.** Two conditioning frames contain disks of different size moving towards each other. The network observes pure noise in the next frame and estimates the target (all input frames are highlighted in red). In this example, the next frame is unambiguous: the light disk on the right should occlude the other. As expected, the denoising estimate (highlighted in green) is a close approximation of the target (but the edges of the disk are blurry). **Middle.** Intermediate steps of the iterative partial denoising procedure, this score-based sampling algorithm uses the same conditional denoiser network as above. The corresponding sampled probable next-frame is highlighted in blue. **Bottom.** Example samples of probable next-frame generated using the iterative partial denoising procedure starting from different random initializations. Each contain sharp occlusion boundaries, all with the light disk on the right occluding the other. With this sampling procedure, the network decides on the occlusion and produces diverse samples that, unlike one step denoising, do not blur the edges.

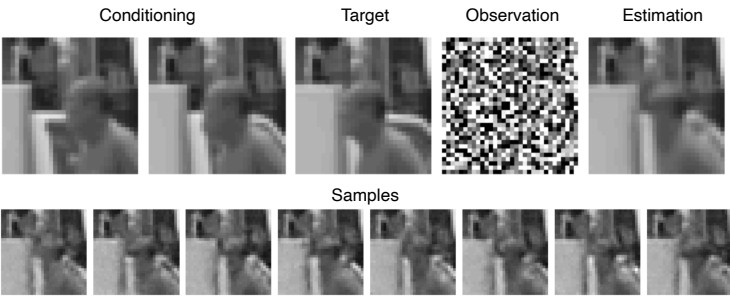

Figure 12: **Samples of probable next-frame.** Sampling from the denoiser with two conditioning frames trained on natural image sequences. **Top.** Example image sequence from the DAVIS test set. The two conditioning frames show a man walking to the left and a car driving to the right in the background. In natural image sequence, the person occludes the car in the next frame. Starting from an observation of pure noise the network estimates a blurry next frame. **Bottom.** Samples starting from different noisy initializations are shown below. They are diverse but their perceptual quality is not very high. In particular the features of the face are lost. Sampling diverse and high quality temporally stable trajectories along the manifold of natural images requires a larger training set and more training iterations.

