# OpenReview forum: "Conditional density estimation for video prediction with score-based models"
_ICLR.cc/2025/Conference — Submitted to ICLR 2025_

### Official Review · Reviewer_rX9D · 2024-10-26

**Soundness:** 4
**Presentation:** 4
**Contribution:** 1
**Rating:** 1
**Confidence:** 2

**Summary:**

The paper proposes a score-based model for conditional density estimation tailored to video prediction. The authors demonstrate that single-step denoising results in blurred predictions, highlighting the necessity of iterative denoising—a standard approach in score matching, diffusion, and flow matching models—to capture sharper, more plausible outcomes. Through a straightforward dataset and network, they further illustrate the distinct contributions of past frames and the current noisy frame in shaping accurate predictions.

**Strengths:**

The paper defines score matching formulation for video prediction in a really accurate way. They also show how adding noise is helpful both in training and inference.

**Weaknesses:**

It’s challenging to identify the novel contribution of this paper, as it seems to be positioned somewhat independently, with limited engagement with prior work on video prediction, particularly with diffusion models. More explicit connections to existing approaches and an analysis of what this model uniquely offers in comparison would strengthen the work.

**Questions:**

Could the authors clarify how their approach to conditional density estimation for video prediction differs from existing score-based or diffusion models?

What specific insights or improvements does this work offer over current state-of-the-art methods in video prediction?

How does the analysis of the network's adaptive behavior contribute to our understanding of video prediction models?

---

> ### Author Response · Authors · 2024-12-04
>
> Thank you for pointing out that the initial manuscript did not discuss enough the relationship of our method with existing literature.
> We have now added a section devoted to related work, and have edited the methods section for clarity.
> Our contribution lies in describing a simple score-based framework for video prediction.
> We claim that it facilitates visualization of the learned representation, casting some light on the adaptivity of trained networks and their ability to exploit spatio-temporal structure in image sequences.
>
> Your question about the analysis of the network's adaptive behavior is perfectly valid.
> The local linear analysis presented here only affords partial access to the adaptive network representation, but it does not elucidate the formation of these representations.
> Further study of deep network implicit biases is necessary, and we believe that the simple framework and visualization methods described here constitute a promising direction.

---

### Official Review · Reviewer_i1kb · 2024-11-02

**Soundness:** 1
**Presentation:** 1
**Contribution:** 1
**Rating:** 1
**Confidence:** 4

**Summary:**

This paper describes a probabilistic formulation for next-frame-prediction.  The paper develops statistical machinery based on learned denoising functions in what appears to be a sampling approach, although the details are vague and hard to follow.  The method is tested on a synthetic, procedural dataset and a small natural image sequence dataset.

**Strengths:**

- The paper motivates a strong problem about the lack of transparency in next-frame generation (or general video generation) papers.
- The base formulation for the next prediction is sound.

**Weaknesses:**

- The development of the technical ideas in the paper are difficult to follow, moving from an initial problem formulation to denoising without any reasonable discussion.   Then the discussion jumps around with relatively basic statistical machinery without giving any details of the actual method.   Better description of what the novelty in the method would be helpful.  More concrete relationship between the current work and the related art is very important to help the reader understand the current method and its contributions.

- The overall approach seems similar to other probabilistic generative methods, even those noted in the paper.

- The evaluation approach is not well described or convincing.  There are not comparisons to component methods or baselines.  It would improve the paper if comparative baselines were included.  The notion of occlusion, which may be useful is introduced seemingly out of nowhere.  It would improve the paper if there is more discussion motivating this evaluation.  However, evaluation should also describe general generation techniques.  The natural image sequences tested are few.  It would improve the paper to demonstrate capability on establish video generation benchmarks with established video generation evaluation protocols, beyond PSNR.  The details of the networks used are unclear.  It would improve the paper to have a thorough network description, even if in the appendix.

- (Minor) Typos exist in the paper; e.g., line 89 focuese

**Questions:**

- The temporal sampling of an image sequence is an artifact of the technology; how does this impact the modeling proposed.  The basic technical premise is disarming: "The next frame in an image sequence is a single event with several possible outcomes..."  The next frame in an image sequence has very very many possible outcomes; it's also not clear what "an event" means (in the context of the earlier part of this bullet).  It would be helpful if the paper included a discussion of the impact of this discrete sampling of continuous time along with a discussion of the potential impact this has on the method's performance.

---

> ### Author Response · Authors · 2024-12-04
>
> Thank you for pointing out that the description of our method was not precise enough in our initial manuscript.
> This section has now been rewritten for clarity and additional information, including detailed algorithms for training and sampling, are provided in the appendix.
>
> You also point out correctly that our initial manuscript did not discuss enough the relationship of our method with existing literature.
> We have now added a section devoted to related work, and could add any missing references in the next version of our manuscript.
> Our approach is indeed generally consistent with the diffusion model formalism and our contribution lies in describing a simple score-based framework for video prediction.
> We claim that it facilitates visualization of the learned representation, casting some light on the adaptivity of trained networks and their ability to exploit spatio-temporal structure in image sequences.
>
> As you noted, we did not present quantitative comparisons to other probabilistic generative methods, but we describe several comparisons: between a pure prediction network and conditional denoising networks, and between conditional denoising networks with varying number of conditioning frames.
> We have also performed additional experiments varying network width and dataset size, these results will be included in the next version of the manuscript, together with more example visualizations of generated image sequences and adaptive filters.
>
> We were too brief in discussing the classic occlusion boundary problem and now discuss this issue in the introduction section to motivate our approach.
> As you mention, digital video are sampled in time, typically at a rate as low as 25 frames per second.
> This heavy subsampling induces temporal aliasing and can sometimes produce noticeable artifacts.
> Remarkably, such sequences of still images are perceive to move by human observers and this phenomenon of apparent motion is at the foundation of all digital video technology.
> Extending the current framework to a continuous time formulation is an intriguing thought that merits further investigation but is beyond the scope of the current study.

---

### Official Review · Reviewer_hMhS · 2024-11-02

**Soundness:** 3
**Presentation:** 2
**Contribution:** 2
**Rating:** 5
**Confidence:** 3

**Summary:**

The manuscript formulates probabilistic forecasting of the next video frame as a generative modeling task. The proposed method allows to recover plausible instances of the next frame by iteratively sampling a denoising deep generative model through Tweedie's formula. In the experiments, the denoising model is a U-Net with up to \tau=2 conditioning frames. The model optimizes the L2 reconstruction loss which corresponds to log p(x|y,c). The experiments have been performed on a synthetic dataset (moving leaves) and DAVIS32.

**Strengths:**

- forecasting future frames in video is an important problem
- generative modelling is an appropriate tool for the task at hand due to ability to account for multimodal future

**Weaknesses:**

- the proposed method appears quite straight-forward and in line with previous recent work in the field
[a] Gabriel Loaiza-Ganem, Brendan Leigh Ross, Luhuan Wu, John P. Cunningham, Jesse C. Cresswell, Anthony L. Caterini. Denoising Deep Generative Models. ICBINB 2022

- the manuscript does not report quantitative comparison with related work in the field; for instance, it would be insightful to report MS-SSIM and LPIPS on Cityscapes and KITTI, as in [b] and references therein.
[b] Yue Wu, Qiang Wen, Qifeng Chen. Optimizing Video Prediction via Video Frame Interpolation. CVPR 2022: 17793-17802.

**Questions:**

- equation (1) is not clear due to index s having two distinct roles (also, it would be helpful to clarify whether s>t is feasible)
- it would be good to clarify whether f(y,c) is E[x|y,c] as in (2) or f(y,c) is E[x|y,c] - y as in (7)

---

> ### Author Response · Authors · 2024-12-04
>
> Thank you for your feedback.
> You point out correctly that our method is generally consistent with the diffusion model formalism.
> Our contribution lies in describing a simple score-based framework for video prediction that facilitates visualization of the learned representation.
> This work casts some light on the adaptivity of trained networks and their ability to exploit spatio-temporal structure in image sequences.
> But the initial manuscript did not discuss enough the relationship of our method with existing literature and we have now added a section devoted to related work.
> Importantly, our proposed method bypasses explicit motion estimation and avoids the classic problems of optic flow at occlusion boundaries.
>
> As you noted, we did not present quantitative comparisons to other video prediction methods.
> Instead of such benchmarking, we want to emphasize the several internal comparisons that we consider: between a pure prediction network and conditional denoising networks, and between conditional denoising networks with varying number of conditioning frames.
> We have also performed additional experiments varying network width and dataset size, these results will be included in the next version of the manuscript.
> We will also provide additional metrics and quantify denoising performance in terms of MSE and SSIM.
>
> Thank you for bringing to our attention the two equations that were not clear in the initial manuscript.
> The new version expresses these points more explicitly and additional details are given in the appendix.

---

### Official Review · Reviewer_8Zrg · 2024-11-09

**Soundness:** 3
**Presentation:** 3
**Contribution:** 4
**Rating:** 6
**Confidence:** 3

**Summary:**

The paper presents a simplified diffusion-based framework for modeling conditional density, with demonstrations in video prediction. The key idea, derived from the empirical Bayes formulation of score-based models, involves learning a denoising function that implicitly infers noise levels and removes noises of arbitrary magnitude from an input. This denoiser yields an estimation of a family of score functions across noise levels. This framework thus removes the time axis in a standard diffusion model, and pursues a direct regression approach, allowing easy analysis of the learned representations. This framework is evaluated on synthetic videos, as well as natural image sequences.

**Strengths:**

The main contribution of this paper is mostly conceptual. The key ideas of (1) employing a direct regression approach for learning score-based models and (2) the sampling strategy from the learned denoising function are both intriguing. It is perhaps a bit surprising to see how they can work even on some toy data.

The proposed framework presents an interesting and significantly simplified alternative to existing diffusion models.

The paper is well-written overall. The results, including a detailed analysis of the learned representations, are elaborated.

**Weaknesses:**

Despite the conceptual novelty, the practicality of the proposed framework is somewhat questionable. Considering the problem of video prediction, it is probably fair to say that the proposed framework provides at best an alternative solution to diffusion models. While diffusion models have demonstrated impressive results for video generation and prediction, the proposed framework is solely demonstrated on “toy” data (small scale, low resolution synthetic and real videos). It is not clear if the proposed framework can scale up to larger datasets or higher resolution videos.

**Questions:**

It will be very helpful to include a discussion that draws the boundary between the proposed framework and well-known diffusion models.

A demonstration of the proposed framework on high resolution videos will help to strengthen the experiments. If this is not possible, a discussion about the scalability might be beneficial.

---

> ### Author Response · Authors · 2024-12-04
>
> Thank you for your feedback.
> As you pointed out, our contribution lies in describing a simple score-based framework for video prediction that facilitates visualization of the learned representation.
> But we had not discussed enough the relationship of our method with existing literature.
> The updated manuscript contains an additional section devoted to related work.
>
> Your question about scalability motivated us to run additional experiments on dataset of varying sizes.
> We observe a systematic reduction of the test loss with the number of training frames, and this trend holds for networks of varying widths.
> These results indicate that our approach can be scaled and will be included in the next version of the manuscript.

---

### Meta-Review · Area_Chair_Aogd · 2024-12-22

**Metareview:**

This paper explores the problem of auto-regressive diffusion-based video generation by casting the problem as denoising a noisy next frame conditioned on clean past frames. Using the connection between the posterior expectation and the score function, the paper derives this next frame prediction problem as approximating the scaled score function from a denoised residual produced by minimizing the mean-squared error in the reverse diffusion process, and proposes a stochastic gradient ascent scheme for the next-frame generation. Experiments are provided on synthetic and real data, showing promise and consistent improvements over easy and hard examples.

**Additional Comments On Reviewer Discussion:**

The paper received four reviews, overall inclined towards rejection. While some of the reviewers liked the simplified regression scheme presented in the paper, all the reviews pointed out important issues with the paper regarding:
1) the paper being mostly conceptual (8Zrg)
2) lack of grounding of the approach with regards to state-of-the-art literature on diffusion models (hMhS, i1kb, rX9D)
3) lack of quantitative comparisons to similar approaches (i1kb, hMhS), and
4) scalability of the approach against state-of-the-art methods (8Zrg).

Authors provided a rebuttal and added a Related Works section in the revised paper. However, no new substantial materials were presented that could convincingly address the issues pointed out by the reviewers.

AC acknowledges the simplified scheme in the paper and the elegance of the formulation when read independently, however also agrees with the reviewers that it is important to derive the technique in the context of other score based formulations already popular in the diffusion-based generation literature to spell out the differences and merits. Given the strong synthesis capabilities of current video diffusion models, it is also inevitable that the paper explores the scalability of the approach to at least some of the datasets typically used for empirical comparisons. As such, the paper has significant shortcomings and thus AC recommends reject.

---

### Decision · Program_Chairs · 2025-01-22

Reject